# T-RippleGNN: Predicting traffic flow through ripple propagation with attentive graph neural networks

Anning Ji[1⊙], Xintao Ma[2⊙]*

1 Department of Traffic Management, Jilin Police College, Changchun, China, 2 School of Management Science and Information Engineering, Jilin University of Finance and Economics, Changchun, China

⊙ These authors contributed equally to this work.
* 119074@jlufe.edu.cn

## Abstract

Recently, accurate traffic flow prediction has become a significant part of intelligent transportation systems, which can not only satisfy citizens' travel need and life satisfaction, but also benefit urban traffic management and control. However, traffic forecasting remains highly challenging because of its complexity in both topology structure and time transformation. Inspired by the propagation idea of graph convolutional networks, we propose ripple-propagation-based attentive graph neural networks for traffic flow prediction (T-RippleGNN). Firstly, we adopt Ripple propagation to capture the topology structure of the traffic spatial model. Then, a GRU-based model is used to explore the traffic model through the timeline. Lastly, those two factors are combined and attention scores are assigned to differentiate their influences on the traffic flow prediction. Furthermore, we evaluate our approach with three real-world traffic datasets. The results show that our approach reduces the prediction errors by approximately 2.24%-62,93% compared with state-of-the-art baselines, and the effectiveness of T-RippleGNN in traffic forecasting is demonstrated.

## 1 Introduction

Traffic management and control have become more challenging with the rapid growth of the number of vehicles during the urbanization process. Effective analysis and prediction of dynamic traffic conditions play a crucial role in the planning and construction of new infrastructure and the management of intelligent transportation systems [1]. However, due to spatial dependence and temporal dependence, traffic flow prediction is becoming increasingly difficult, and traditional methods are not applicable.

Firstly, the change in traffic conditions is affected by the topology structure of the traffic. Roads are part of a complex web where congestion in one area can quickly impact surrounding regions owing to the interconnected nature of the roads [2]. When the traffic condition changes or congestion occurs at the upstream point of the road,

**Data availability statement:** The underlying data from this study can be found at https://github.com/helenma27/TRippleGnn.

**Funding:** X. Ma has received the funding Jilin Provincial Department of Education Science and Technology Research Project(JJKH20240199KJ). The funders had no role in study design, data collection and analysis, decision to publish, or preparation of the manuscript.

**Competing interests:** No authors have competing interests.

other downstream points along the traffic flow direction would also be influenced by the transfer effect [3]. As shown in Fig 1, the volume of point A gradually affects the adjacent roads differently with the distance of neighbors. The impact moves further along the network from points B and C to D.

Secondly, traffic flow prediction has the temporal characteristic, where the traffic condition changes dynamically over time, exhibiting long-term periodicity and trend. However, the uncertainties in the case of short periods, including the temporal evolution of the traffic flow caused by incidents, are difficult to forecast [4].

Traditional approaches generally deal with the temporal characteristic, and they are classified into two categories, namely statistical methods such as Auto-Regressive and Moving Average (ARMA)[5,6], Support Vector Regression (SVR) [7], and Kalman filtering method [8,9], and deep learning methods, like Deep Recurrent Neural Networks (RNN) [10] and its successors Long Short Term Memory (LSTM) networks [11]. However, these methods encounter problems when it comes to dynamic real traffic conditions because of their incapability of handling spatial information of roads. Therefore, graph neural networks (GNN) have lightened the researchers to mine the topology relation of the traffic roads [12], including T-GCN [2], GaAn [13], and traffic-GGNN [14]. The key idea of GNN is to explore the non-Euclidian correlations, especially the irregularity in the traffic topology.

These GNN-based traffic flow prediction (FTP) systems have proved to be effective and show superior performance because of the consideration of traffic topology. However, it is argued that two problems remain unsolved. On the one hand, GNN-based models capture the spatial relations by transforming and aggregating information through the edges of the road network, yet fail to model the dynamic characteristic of the traffic influence from one road to another. As a result, it is crucial to model the dynamic traffic propagation, especially the heavy traffic load impact, which transfers to neighbors from near to far. On the other hand, they normally apply

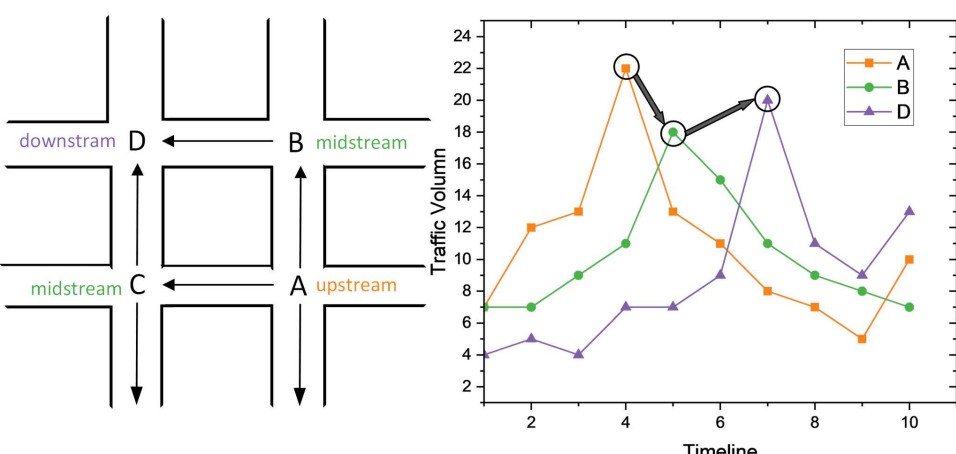

**Fig 1. Spatial dependency is affected by the topology structure of traffic networks.** Due to the impact between adjacent roads, the short-term traffic flow is transferred from upstream A to downstream D from near to far.

GNN to explore the spatial information and then RNN for temporal information, aggregating those two characteristics with a multilayer perceptron (MLP) after two-aspect exploration is accomplished. We consider that this kind of fusion neglects the inner relation between spatial and temporal features, thus easily falling into partial optimum with respect to either spatial or temporal factors.

In order to overcome those limitations, we propose Ripplenet attentive graph neural networks to predict traffic flow (T-RippleGNN), which utilizes Ripple propagation to capture the traffic spatial information and the gated recurrent unit (GRU) to explore the traffic through the timeline. The key idea of traffic topology exploration of the GNN-based model is information propagation, with which the structure of the road network can be constructed. Inspired by Ripplenet[15], we believe that the traffic flow propagation is similar to actual ripples created by raindrops propagating on the water, specifically the heavy traffic volume could affect the neighbors gradually through the propagation. Secondly, a GRU-based module is used to extract the temporal correlations after each propagation step. Finally, those two factors are combined with attention mechanisms to assign different weights to different locations regarding spatial and temporal influence on traffic flow. Our codes are available at https://github.com/helenma27/TRippleGnn. In summary, the main contributions of this study are as follows:

- The concept of Ripplenet is employed to mine the topology structure of traffic networks by propagation traffic flow conditions like ripples, which can effectively explore the high-order correlations among traffic roads.

- The temporal and spatial features are integrated after each time step, which could explore the global context of traffic dynamic topology through the timeline. Then, we reweight the importance of the influences of road neighbors through the timeline.

- Our framework is evaluated with extensive experiments on three real-world datasets, SZ-taxi, Los-loop, and PEMS-BAY. The results demonstrate that our framework achieves a better performance compared with other state-of-the-art algorithms.

The rest of this paper is organized as follows: firstly, some related works regarding traffic forecasting and GNN-based models are introduced in Section 2. Then, some notations are illustrated, and our framework is explained in Section 3. Following that, the experiments are demonstrated with the evaluation metrics, and the results are analyzed in Section 4. Finally, our work is concluded, and some future work is shown in Section 5.

## 2 Related work

Traffic forecasting has drawn increasing attention recently because of its fundamental applications in transportation aspects, including traffic planning and reducing traffic jams. However, efficient and accurate traffic forecasting remains challenging since the number of vehicles is continuously growing with the rapid urbanization process, and the dynamic pattern in traffic prediction from both spatial and temporal point of view.

Existing traffic forecasting can fall into two categories: knowledge-driven approaches and data-driven approaches. The former methods often need superior and professional knowledge, as they tend to precisely model the traffic system regarding traffic volume, speed, and density. The typical models include the queuing theory method and the "traffic velocity" method [16]. However, the traffic is not only impacted by the factors shown before, which is non-trivial to build a model that absorbs all factors.

The latter method uses historical traffic data to predict future traffic states, and they can also be classified into two groups: traditional methods and deep learning methods. The traditional methods involve statistical based (such as ARIMA [5] and vector autoregressive [17]) and machine learning based (such as K-nearest neighbor [18] and support vector machine [19]) ones. Yang et al. proposed a hybrid model that integrated the extreme learning machine and an improved whale optimizer to enhance the robustness of the prediction [20].However, the proliferation of data and the complexity of

 

road networks relegate traditional methods to the background. Thus, the rise of deep learning methods has given traffic forecasting another possibility to improve the efficiency and accuracy of prediction. RNN [21,22] and its variants including LSTM[23] and GRU[24] can be directly used for traffic forecasting owing to their capability to efficiently deal with sequence information such as traffic data. Combined with attention mechanisms, RNN can adaptively select the most relevant hidden states in the history sequence to predict the current road states [25–27]. For instance, Zheng et al. put forward Gman that integrates attention mechanisms with the encoder, capturing the non-linear correlations between different time slices [25]. TGN [28] provides a general framework that operates on dynamic graphs represented as event sequences and a novel training strategy that enables the model to learn from the sequential nature of the data while maintaining efficient parallel processing capabilities. Yang et al. pointed out that M-R-AR-BiGR [29] was combined with the traditional methods and bi-directional gate recurrent units to capture the temporal characteristics and improve the robustness.

Furthermore, the ability of graph neural networks (GNN) to capture the temporal-spatial dependency and represent the correlations for the non-Euclidean spaces has attracted the attention of researchers since 2019. It can be categorized into recurrent graph neural networks (RecGNN) [30], graph convolutional networks (GCN) [31], graph autoencoders (GAE)[32], and the Spatial-Temporal GNN [33]. Currently, the most widely used GNN for traffic forecasting is the GCN, which propagates the node states to the neighbors and then updates them, so that the whole graph topology is grasped [34–36]. Additionally, recent research has introduced various GCN and RNN models that can effectively capture the complex interdependencies inherent in spatial and temporal data. T-GCN [3] integrates GNN with GRU to capture both the topology structure of the traffic network and traffic dynamic patterns. STFGNN [37] enhances traffic forecasting by fusing temporal and spatial graphs with GCN and gated convolutions, with the capability to effectively address long sequence data. DyHSL [38] improves the forecasting with hypergraphs using HGNN for dynamics and interactive convolutions for spatial-temporal relations, which can capture higher-order interactions but at the cost of increased computational complexity and scalability issues. STGNN [39] integrates a novel graph neural layer, a recurrent layer and a transformer layer to capture the temporal and spatial dependency, which is more feasible for long period traffic speed prediction.

However, our framework differs from the above literature since the propagation idea is followed to capture the traffic network topology gradually, especially in the heavy traffic states. Ripple propagation is combined with GRU in an iterative manner, where spatial propagation and temporal modeling are deeply intertwined. This enables T-RippleGNN to not only extract spatial features more effectively but also refine them iteratively based on temporal context, leading to a more comprehensive representation of traffic patterns. Both STFGNN and STGNN employ graph convolutional networks to model spatial dependencies, but their reliance on predefined static graphs limits their ability to adapt to dynamic traffic conditions and long-term prediction. In contrast, our model leverages Ripple to propagate traffic influences adaptively across multi-hop neighbors, capturing not only local spatial correlations but also global traffic patterns, especially under heavy traffic conditions.

## 3 Proposed work

Our framework T-RippleGNN is shown in Fig 2. The left part is the whole architecture of our framework, which contains spatial and temporal feature extraction, and a final attention mechanism to reweight the location influence. The traffic flow profile at each time step is taken as input. It is through each Ripple propagation and GRU module as shown at the right part of the figure. Then, the attention mechanism is displayed in the upper part of the figure, to differentiate the location weight according to both the spatial and the temporal influences on the final traffic flow prediction. One thing to mention is that the parameters of each GRU are shared through the whole framework.

At first, some necessary notations and formulations are shown in subsection 3.1. Then, our framework module is explained by the modules, namely the Ripple module in subsection 3.2, the GRU in subsection 3.3, and prediction units in subsection 3.4. Finally, the learning algorithm will be discussed in subsection 3.5.

### 3.1 Formulation

First, some notations and formulations are introduced.

- **Traffic network**: The traffic network can be regarded as graphs $G = (V, E)$ with $N$ traffic sensor nodes $V = \{v_1, v_2, \ldots, v_n\}$ and edges $e \in E$ between nodes $(v_1, v_2) \in E$. Although in reality, the roads are directed, taking into account that the traffic congestion is bi-directionally propagated to upstream and downstream roads, $G$ is taken as an undirected graph, as the propagation is bi-directional [40]. The historical traffic conditions are denoted as $X = \{x_N^1, x_N^2, \ldots, x_N^T\} \in \mathbb{R}^{N \times T}$ of the traffic network. The traffic conditions can be traffic flow, speed, density, and so on.

- **Traffic flow prediction**: Given the traffic network $G = (V, E)$ and the historical traffic flow profiles $X = \{x_N^1, x_N^2, \ldots, x_N^T\} \in \mathbb{R}^{N \times T}$, the traffic flow prediction problem is to learn a model $f$ for calculation in the next time $\tau$, $\hat{Y} = \{\hat{Y}_N^{t+1}, \hat{Y}_N^{t+2}, \ldots, \hat{Y}_N^{t+\tau}\}$.

### 3.2 The Ripple module

In this module, the Ripple propagation is implemented to spread the traffic conditions over the connecting road nodes and to explore the spatial relevance of the traffic network. In this study, we argye that the traffic profiles can be propagated through the edges as the raindrops fall into the river and provoke ripples, in which multiple ripples superpose to form a resultant preference distribution of the traffic condition.

We first define the concept of the ripple set regarding to the traffic network as follows:

$$\varepsilon_n^k = \{v_t | (v_h, e, v_d) \in G, \quad v_h \in \varepsilon_n^{k-1}\} \tag{1}$$

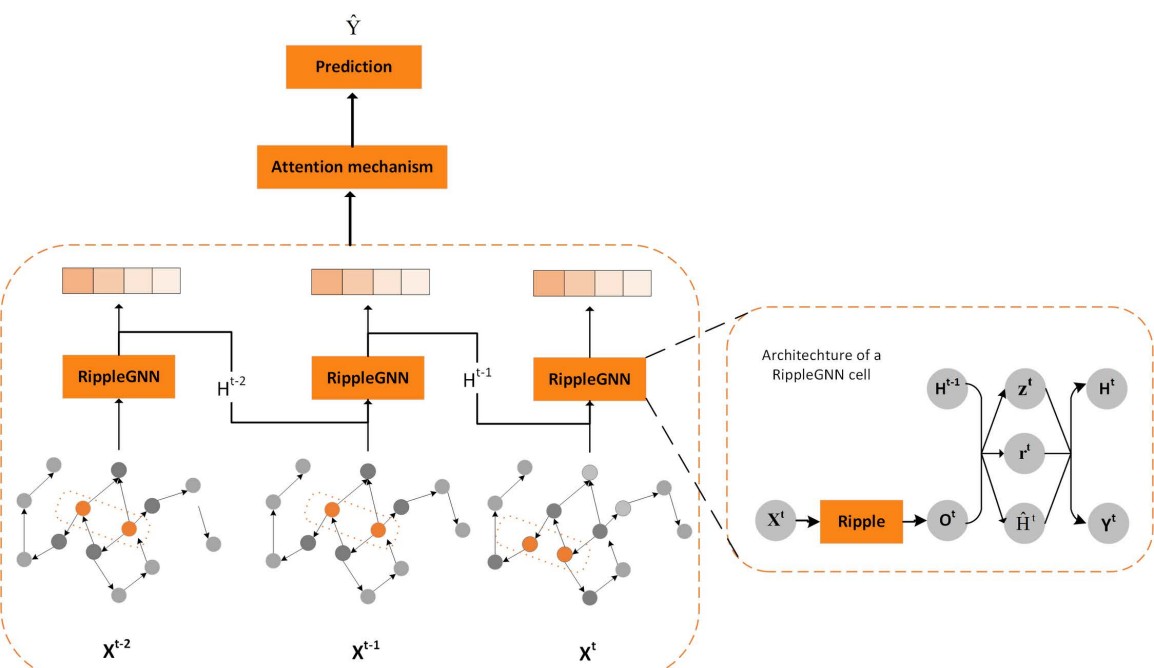

**Fig 2. The framework of T-RippleGNN: The right part represents the specific architecture of a RippleGNN cell, where Ripple means Ripple propagation for the input traffic conditions.** Then GRU is applied after Ripple propagation.

where $k = \{1, 2, \ldots, H\}$; $H$ is the hop number, and $\varepsilon_n^0 = \{n\}$ refers to the initial point $n$. Therefore, the ripple set of a traffic node can be defined as triples with the start node to the connected node neighbors belonging to $(k-1)$-hop relevant nodes.

$$S_n^k = \{(v_h, e, v_d) | (v_h, e, v_d) \in G, \quad v_h \in \varepsilon_n^{k-1}\} \tag{2}$$

As shown in Fig 3, the traffic condition can be spread through the ripple set at each hop, and then propagates along the links in the traffic networks, from near to far. Considering the traffic congestion influence attenuated with the increase of the hop number $k$, as the ripple decays, Fig 3 also shows different grey-level with the decreasing relatedness between starting nodes with neighbors.

The propagation process can be illustrated as follows: Start points are randomly chosen as seeds $\varepsilon_n^0$, and then the k-hop ripple set can be calculated by Eq. 2. Starting from the 1-hop ripple set $\varepsilon_n^1$, we assign each neighbor a relevance probability that the traffic condition can be propagated from $v_h$ to $v_t$ through the edge $e \in E$ as displayed in Eq.3.

$$p_i = \frac{exp(v_h^T v_{d_i})}{\sum_{(v_h, e, v_d) \in S_n^1} exp(v_h^T v_d)} \tag{3}$$

where $s$ represents the embedding dimension, and $v_d \in \mathbb{R}^s$. This probability $p_i$ is calculated based on the similarity between the embeddings of the seed node and its neighbors, revealing the impact strength between the connected traffic nodes.

Using the relevance probabilities, we aggregate the traffic conditions of the neighbors to compute the first hop response $(o_n^1)$ for the seed node, as displayed in Fig 3(b). This response represents the traffic state of the seed node after incorporating the influence of its immediate neighbors.

$$o_n^1 = \sum_{(v_{h_i}, e_i, v_{d_i}) \in S_n^1} p_i v_{d_i} \tag{4}$$

Thus, the first response $o_n^1$ of the traffic profile in node $n$ can be obtained. The propagation continues iteratively for k-hops. At each step, the response from the previous hop $(o_n^{k-1})$ is used as the new seed to compute the next response $(o_n^k)$, by iteratively replacing $v_h$ by the previous response $o_n^{k-1}$ in Eq.3 with the node ripple sets $S_n^h$ for $k = 1, 2 \ldots K$. This allows the

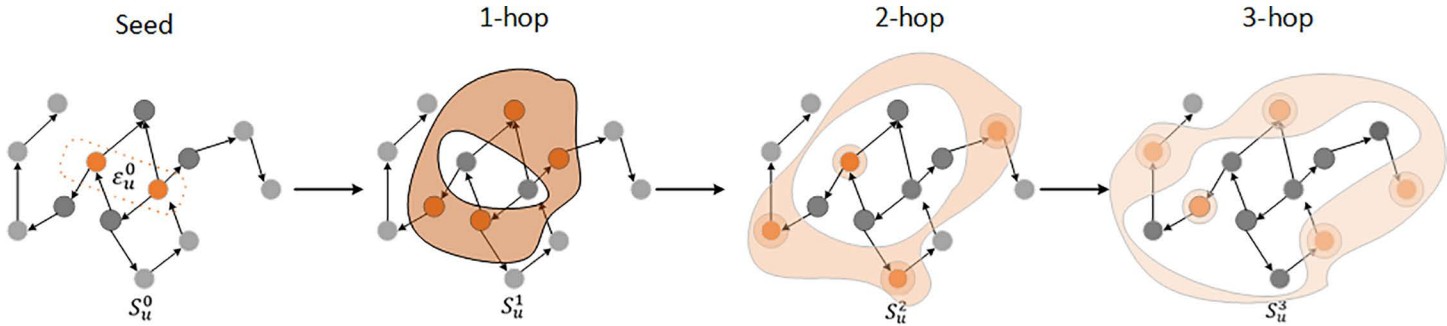

**Fig 3. The water wave shape of Ripple propagation procedure: The start points $\varepsilon_n^0$ are randomly selected.** The traffic conditions propagate to its immediate neighbors, forming the 1-hop ripple set $(\varepsilon_n^1)$. During this step, the feature embeddings of the seed nodes are transmitted to their neighbors. The strength of this influence is represented by the intensity of the circle's color, with darker shades indicating stronger influence. This propagation continues hop-by-hop until the whole structure is propagated or the limit hop number is reached.

model to capture traffic influences from increasingly distant nodes, reflecting the dynamic and multi-scale nature of traffic networks. Then, the embedding of traffic node $n$ can be considered as the combination of all responses:

$$o_n = o_n^1 + o_n^2 + \ldots + o_n^K \tag{5}$$

To reduce the computational complexity, the hop number threshold is set to a small number, while the traffic condition can still be propagated to the whole network. The reason is that the relevant nodes or neighbours are still undercover as the ripples overlap during different hop propagations [15]. In other words, a traffic node can be reached from multiple paths in the traffic networks.

Therefore, by iteratively calculating, the traffic flow embedding vectors can be acquired and formalized as $O^t = \{o_1^t, o_2^t, \ldots o_N^t\}$, where N denotes the number of traffic nodes.

### 3.3 The GRU module

To capture the temporal dependency, the GRU is adopted to deal with the time-sequence data. As shown in subsection 3.2, the traffic node embedding for the specific time step can be calculated by Ripple propagation. $O_n^t$ is defined as the traffic condition of node $n$ at time $t$. To corporate the spatial relations while processing the time sequence, we follow the work [14,41] that applies GRU to obtain the traffic flow embedding. Given the input obtained from the Ripple propagation $O^t$, and the hidden representations from the previous time step $H^t$, the operation of GRU can be expressed as follows:

$$z^t = \sigma_z(W_z O^t + U_z H^{t-1} + b_z) \tag{6}$$

$$r^t = \sigma_r(W_r O^t + U_r H^{t-1} + b_r) \tag{7}$$

$$\hat{H}^t = tanh(W_h O^t + U_h \left(r_t \odot \hat{H}^{t-1}\right) + b_h) \tag{8}$$

$$H^t = \hat{H}^t \odot z^t + \hat{H}^{t-1} \odot (1 - z^t) \tag{9}$$

where $\odot$ represents the element-wise multiplication and $\sigma$ is the sigmoid function; $W_z$, $W_r$, $W_h$, $b_z$, $b_r$ and $b_h$ are the trainable weights and bias of GRU. $z^t$ and $r^t$ are the update gate and the reset gate respectively. The reset gate determines how to combine new input information with previous memory, while the update gate defines the amount of previous memory saved to the current time step.

### 3.4 The prediction module

In this module, the importance of each node in the traffic condition propagation is reevaluated after the temporal characteristic extraction, which integrates the spatial and temporal dependency. We apply the self-attention mechanism which only takes the sequence $H^t$ as input and calculates the attention score.

$$a = \frac{exp(WH^T(H^T)^{-\frac{1}{2}})}{\sum_N exp(WH^T(H^T)^{-\frac{1}{2}})} \tag{10}$$

$$H_{out}^T = a \odot H^T \tag{11}$$

where $W$ is the trainable parameter to balance the spatial influence inside $H^t$.

Then, $H^T_{out}$ can be fed into a multi-layer perceptron (MLP) to produce the final traffic prediction:

$$\hat{Y} = W_p H^T_{out} + b_p \tag{12}$$

where $W_p$ and $b_p$ are the trainable weights and bias respectively.

### 3.5 Learning algorithm

The loss function of our framework is defined as follows:

$$L = L_{PRE} + L_{REG} \tag{13}$$

RMSE is used as the prediction loss function.

$$L_{PRE} = \sqrt{\frac{1}{s} \sum_{i=1}^{s} (y_i - \hat{y}_i)^2} \tag{14}$$

The last loss term is the regularization term that prevents overfitting, by taking all the parameters in the framework into consideration.

$$L_{REG} = \lambda_2 \|W\|_2^2 \tag{15}$$

In conclusion, the whole learning algorithm is shown in Algorithm 1: The training process contains two procedures: from line 3–5, the spatial dependency is explored with Ripple propagation; and line 7–8 represents the training of GRU together with the attention unit:

**Algorithm 1. The learning algorithm of T-RippleGNN.**

```
Input: traffic condition X = {x¹_N, x²_N, …, x^T_N}, traffic graph G = (V, E).
Output: Prediction X_pred = {x^{t+1}_N, x^{t+2}_N, …, x^{t+τ}_N}
1: Initialize the parameters, the initial node n
2: For i=1,…, T do
3: For k hops do
4: Sample minibatch of positive and negative edges from G
5: Calculate the gradients on the minibatch by back-propagation according to Eq. 3–5
6: end for
7: Calculate the gradients on the minibatch by back-propagation according to Eq. 6–12
8: Update the parameters by gradient descent with learning rate η
9: end for
```

## 4 Experiments

In this section, the performance of our framework on three real-world datasets is shown, and the results with state-of-the-art baselines are compared, when parameter sensitivity experiments are conducted.

### 4.1 Dataset

Three common real-world datasets: SZ-taxi, Los-loop, and PEMS-BAY are implemented, and are widely used in the field of traffic prediction. The details are shown below:

- **SZ-taxi:** This is the taxi trajectory of Shenzhen from Jan. 1 to Jan. 31, 2015. It contains 156 major roads of Luohu District as the study area. The speed of traffic on each road is calculated every 15 minutes.

- **Los-loop:** This dataset is collected on the highway of Los Angeles County in real time by loop detectors. It includes 207 sensors, and its traffic speed is collected from 3/1/2012 to 3/7/2012. The traffic speed data is aggregated every 5 minutes.

- **PEMS-BAY**: It contains 6 months of statistics on traffic speed, ranging from 1/1/2017 to 6/30/2017, including 325 sensors in the Bay area.

Totally 80% of the datasets are selected as the training set and the remaining as the test set. The detail statistics are shown in Table 1. The data of the previous 60 minutes is used to predict the speed for the next 15, 30, and 60 minutes.

## 4.2 Baselines

Our results are compared with some state-of-the-art algorithms, which can be classified into traditional statistical methods, deep learning-based models, and GNN-based models. They are listed as follows:

**ARIMA**[5]: It is a widely-used in time series analysis, which is a classical model for predicting the future states in the timeline.

**SVR**[7]: It uses a support vector machine to process regression on the traffic data sequence, which is also a traditional statistical method.

**FNN**[42]: The Feed forward Neural network can be directly employed to deal with traffic sequences, but only model the temporal dependencies.

**GRU**[41]: It is RNN's variant model, which can effectively deal with traffic sequences regarding temporal dependencies.

**T-GCN**[2]: Temporal GCN combines the graph convolutional network with the GRU for traffic forecasting.

**STGNN**[39]: It combines a recurrent layer and a Transformer layer for temporal dependency, and a Graph neural network layer to process the spatial dependency.

## 4.3 Experiment setup

Those models are evaluated with three evaluation metrics: RMSE, MAP, and MAPE, where the smaller values indicate better prediction performance.

(1) Root Mean Squared Error (RMSE):

$$RMSE = \sqrt{\frac{1}{N} \sum_{i=1}^{N} (y_i - \hat{y}_i)^2}$$

where N is the number of the samples; $y_i$ and $\hat{y}_i$ denote the actual traffic and the predicted values respectively.

**Table 1. The details of the three real-world datasets.**

| Datasets | SZ-taxi | Los-loop | PEMS-BAY |
|---|---|---|---|
| Nodes | 156 | 207 | 325 |
| Edges | 532 | 2833 | 2369 |
| Time Steps | 2976 | 2016 | 52116 |
| Mean | 12.2 | 58.9 | 62.6 |

(2) Mean Absolute Error (MAE):

$$MAE = \frac{1}{N} \sum_{i=1}^{N} |y_i - \hat{y}_i|$$

It is the average of the absolute errors.

(3) Mean absolute percentage error (MAPE):

$$MAPE = \frac{1}{N} \sum_{i=1}^{N} \left| \frac{y_i - \hat{y}_i}{y_i} \right| \times 100\%$$

Besides, the ripple hop number of 3 is employed for SZ-taxi and Los-loop, and hop number of 2 is for PEMS-BAY. A larger number of hops hardly improves performance but results in heavier computational complexity. In addition, the embedding dimension and the learning rate are set as 32 and 0.001, respectively.

## 4.4 Results and discussions

In this subsection, the results of comparison with other baselines are presented, and then some parameter sensitivity experiments are discussed.

 **Overall comparison.** The comparison results with the baselines regarding the traffic prediction are illustrated in Table 2–Table 4. Then, the following observations can be obtained:

- Our framework outperforms other baseline models almost on all the long-term and short-term settings. For example, for the 15-minute traffic prediction on the PEMS-BAY dataset, compared with the best result of STGNN, the values of RMSE, MAE, and MAPE have been decreased by 2.24%, 3.52%, and 3.36% respectively. For the long-term prediction of 60 minutes, compared with the best result among baselines on the dataset PEMS-BAY, the performance is increased by 5.05%, 7.60%, and 3.52% for the three evaluation metrics respectively. The results demonstrate the effectiveness of our proposed model, especially on the dataset with complex topology. The reason is that Ripple propagation can further acquire inner correlation among roads, as the traffic information flows through the network. Moreover, our model uses a recurrent unit directly after each time propagation rather than a spatial-temporal separate extraction and a delayed fusion. Finally, an attention mechanism is applied to reweight the impact of different roads.

**Table 2. Performance comparison of T-RippleGNN and other baseline methods on SZ-Taxi dataset.**

| Model | 15min | | | 30min | | | 60min | | |
|---|---|---|---|---|---|---|---|---|---|
| | RMSE | MAE | MAPE | RMSE | MAE | MAPE | RMSE | MAE | MAPE |
| ARIMA | 6.804 | 4.680 | 29.73% | 6.804 | 4.680 | 29.72% | 6.796 | 4.676 | 29.71% |
| SVR | 4.152 | **2.627** | **21.09%** | 4.168 | 2.689 | 21.83% | 4.220 | 2.776 | 22.76% |
| FNN | 6.534 | 3.347 | 29.29% | 4.160 | 2.822 | 23.48% | 4.201 | 2.887 | 22.89% |
| GRU | 6.298 | 3.114 | 28.11% | 4.098 | 2.898 | 22.58% | 4.187 | 2.829 | 22.67% |
| T-GCN | 4.137 | 2.825 | 23.00% | 4.159 | 2.845 | 23.49% | 4.200 | 2.802 | 23.25% |
| STGNN | 4.072 | 2,734 | 22.16% | 4.076 | 2.786 | 22.29% | 4.087 | 2.770 | 22.83% |
| **T-Ripple GNN** | **4.041** | 2.658 | 21.79% | **4.060** | **2.683** | **21.80%** | **4.069** | **2.740** | **22.12%** |

Table 3. Performance comparison of T-RippleGNN and other baseline methods on Los-loop dataset.

| Model | 15min | | | 30min | | | 60min | | |
|---|---|---|---|---|---|---|---|---|---|
| | RMSE | MAE | MAPE | RMSE | MAE | MAPE | RMSE | MAE | MAPE |
| ARIMA | 10.054 | 7.706 | 20.84% | 10.059 | 7.711 | 20.86% | 10.066 | 7.717 | 20.90% |
| SVR | 5.651 | 3.864 | 9.33% | 6.734 | 4.647 | 11.51% | 8.027 | 4.320 | 14.65% |
| FNN | 5.244 | 3.501 | 9.03% | 6.148 | 3.780 | 9.88% | 7.885 | 4.431 | 13.89% |
| GRU | 5.217 | 3.344 | 8.41% | 6.123 | 3.771 | 9.90% | 7.466 | 4.415 | 13.06% |
| T-GCN | 5.209 | 3.275 | 8.34% | 6.117 | 3.734 | 9.76% | 7.266 | 4.403 | 12.31% |
| STGNN | 5.081 | 2.954 | 7.86% | 6.034 | 3.489 | 9.26% | 7.170 | 4.114 | 12.00% |
| **T-Ripple GNN** | **4.958** | **2.850** | **7.39%** | **6.001** | **3.382** | **9.14%** | **7.108** | **4.085** | **11.51%** |

Table 4. Performance comparison of T-RippleGNN and other baseline methods on PEMS-BAY dataset.

| Model | 15min | | | 30min | | | 60min | | |
|---|---|---|---|---|---|---|---|---|---|
| | RMSE | MAE | MAPE | RMSE | MAE | MAPE | RMSE | MAE | MAPE |
| ARIMA | 6.254 | 4.215 | 10.42% | 6.592 | 3.899 | 8.16% | 6.643 | 6.046 | 9.23% |
| SVR | 4.905 | 4.530 | 7.57% | 5.617 | 4.850 | 8.60% | 6.610 | 5.520 | 10.10% |
| FNN | 4.420 | 2.211 | 5.19% | 4.632 | 2.391 | 5.43% | 4.987 | 2.468 | 5.89% |
| GRU | 4.192 | 2.054 | 4.85% | 4.557 | 2.200 | 5.71% | 4.962 | 2.374 | 5.70% |
| T-GCN | 2.723 | 1.482 | 3.11% | 3.314 | 1.777 | 3.70% | 3.965 | 2.108 | 4.571% |
| STGNN | 2.430 | 1.189 | 2.38% | 3.271 | 1.466 | 3.09% | 4.201 | 2.039 | 4.55% |
| **T-Ripple GNN** | **2.318** | **1.179** | **2.30%** | **3.017** | **1.424** | **3.05%** | **3.989** | **1.884** | **4.39%** |

- Traditional statistical methods including ARIMA and SVR perform well for a short-term prediction, especially for a 15-minute test on the SZ-Taxi. SVR outperforms the deep-learning methods and even our proposed model. However, their weakness lies in long-term prediction and complex road networks. For all the datasets, ARIMA and SVR have a higher RMSE and MAE compared with other RNN-based and GNN-based approaches, dealing with 45-minute and 60-minute predictions.

- Recently proposed GNN-based models perform better than traditional deep learning methods such as FNN and GRU. They achieve noticeable improvements on the datasets SZ-Taxi and PEMS-BAY. One reason is that they adopt graph neural networks to process the spatial dependency. Combined with temporal information, they can discover more correlation among data, which is consistent with the previous observations [12,42]. STGNN has a better performance compared with T-GCN because it explores both global and local temporal dependencies, preventing the local minimum problem.

**Parameter variation.** Firstly, our framework is evaluated on the parameter sensitivity by varying the hop number from 1 to 5. The results are shown in Figs 4–6. Obviously, our framework achieves the best when  for the SZ-Taxi and Los-loop datasets, and  for the PEMS-BAY dataset. The reason is that a higher hop number of Ripple propagation module introduces noise during training, especially when low-order connectivity already includes those neighbors also covered by high-order propagation. It is of note that with , the performance is the worst as the propagation has not taken place, thus the model containing only the information of the time line, which is almost equivalent to GRU.

Moreover, the impact of the higher hop number on the dataset PEMS-BAY influences worse than on the dataset SZ-taxi, as the more complex the graph is, more noise is introduced by overlapped propagation traces. For instance, when $h = 5$, RMSE = 4.960 for 60-minute prediction, with an increase of 19.58% compared with $h = 5$ on the dataset

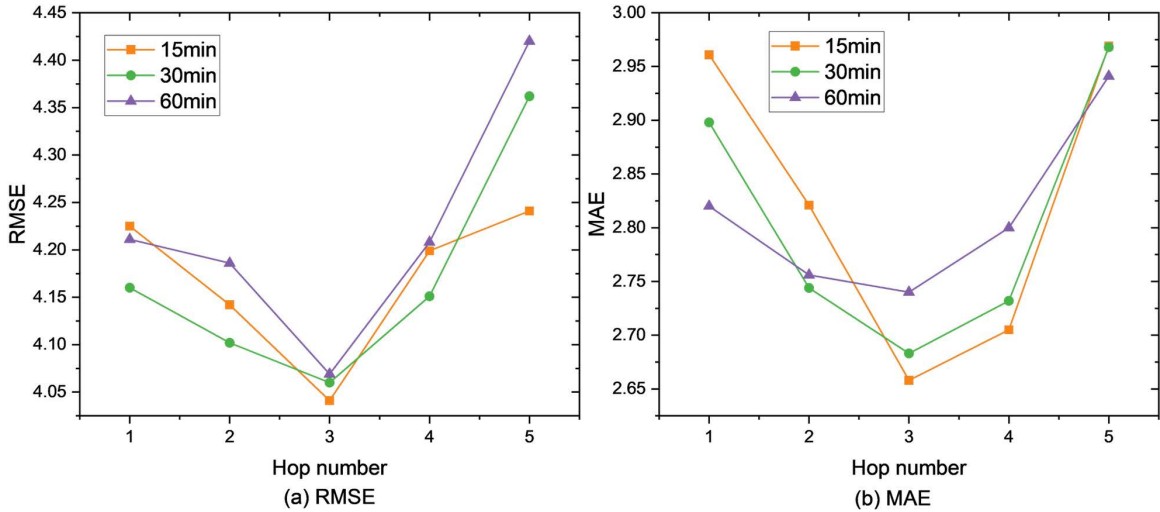

**Fig 4. Prediction metrics for SZ-taxi.**

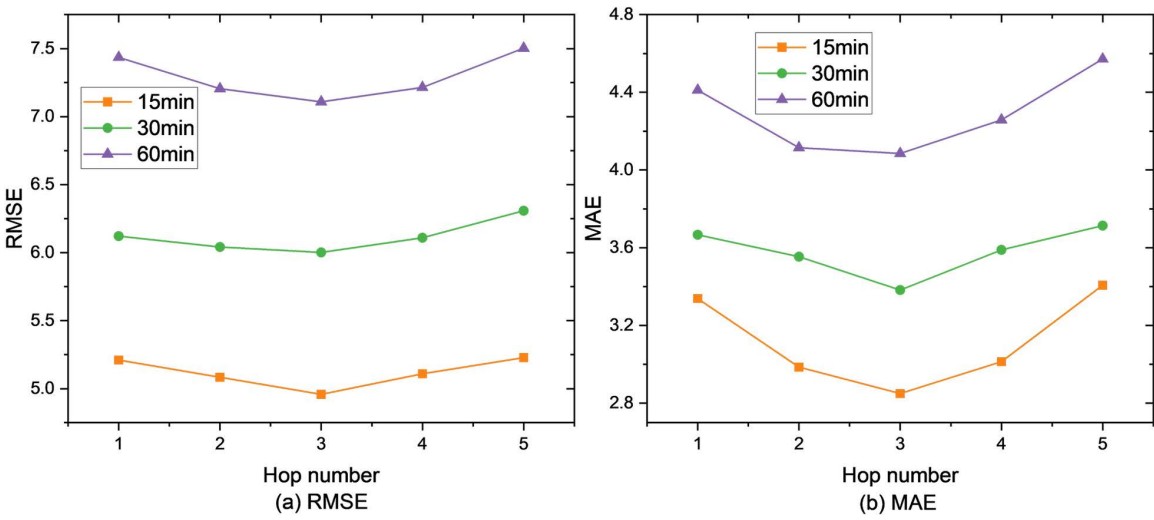

**Fig 5. Prediction metrics for Los-loop.**

PEMS-BAY. However, when $h = 5$, RMSE = 4.42 for 60-minute prediction, increasing by 8.62% compared with $h = 5$ on the dataset SZ-taxi. Therefore, the hop number should be carefully chosen: In simpler graphs (e.g., SZ-Taxi), higher hop numbers ($h > 3$) introduce less noise, as the propagation traces are less likely to overlap. However, in more complex graphs (e.g., PEMS-BAY), higher hop numbers lead to significant performance degradation due to overlapping propagation paths and increased noise. Thus, in such cases, $h \leq 3$ is a better parameter option.

The size of the traffic condition is varied in each hop to further investigate the robustness of Ripple propagation. The result is shown in Fig 7, and it can be seen that the best embedding size occurs at 32. The RMSE first decreases as the size becomes larger, and then increases when reaching the best result, since a size of 32 is enough to carry the information for propagating to neighbor nodes. In more generalized cases, the embedding size is 32, and it achieves the best performance in the large-scale dataset PEMS-BAY, striking a balance between expressiveness and computational efficiency [15].

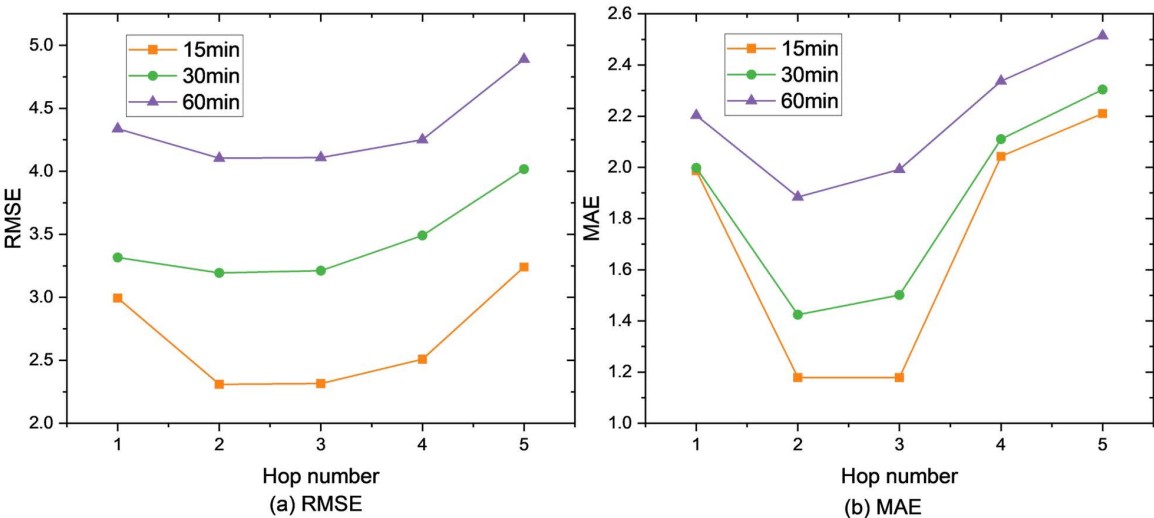

**Fig 6. Prediction metrics for PEMS-BAY.**

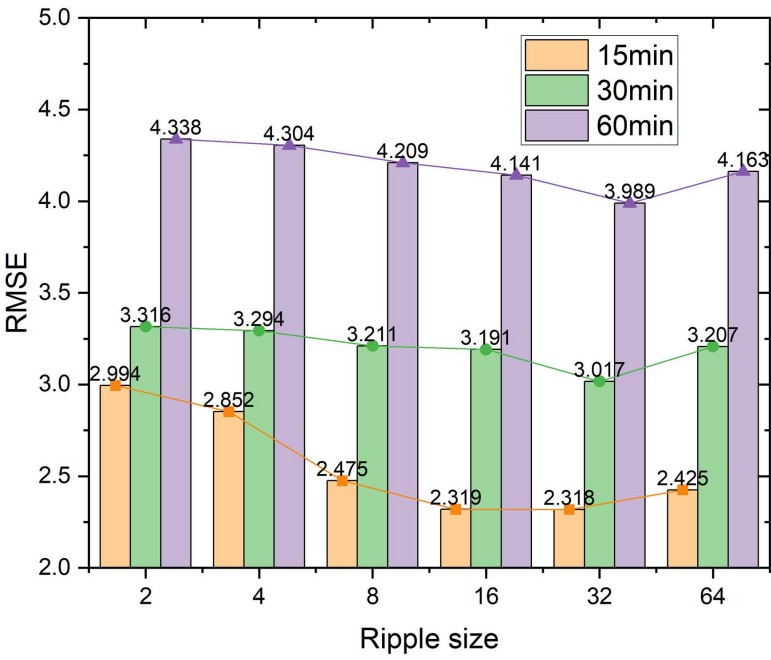

**Fig 7. RMSE on the dataset PEMS-BAY with different ripple sizes.**

In summary, while T-RippleGNN demonstrates effective performance in traffic flow prediction, it also has some limitations. Although the multi-hop propagation mechanism is effective for capturing spatial dependencies, it incurs increased computational overheads when applied to datasets with extremely large graph sizes. Specifically, the iterative aggregation of h-hop neighbors leads to a time complexity proportional to $O(k \times |E|)$, where $|E|$ is the number of edges. Therefore, the choice of hyper parameters, particularly the maximum hop number $h$, should be adapted according to the graph complexity. While larger $h$ values enable the capture of long-range spatial dependencies, they also introduce noise from

irrelevant distant nodes and increase the risk of overfitting. Our ablation studies above reveal that performance plateaus when $h = 3$ for most tested scenarios, reaching a balance between computational cost and efficiency.

## 5 Conclusion

To conclude, in this study, we propose T-RippleGNN for traffic forecasting. It explores the spatial correlation by Ripple propagation, and then investigates the temporal characteristic by GRU. Then, an attention mechanism is applied to reevaluate the importance of locations regarding spatial-temporal fusion. For experiment, our framework is tested compared with the state-of-the-art baselines. The results indicate that our framework improves the prediction quality and efficiency.

For future work, we plan to investigate other types of traffic conditions, such as traffic density or dynamic events. Instead of fixed hop numbers, we would also develop a dynamic mechanism to adjust the hyper parameters based on the real-time traffic congestion. Moreover, we would also further analyze the exploration of temporal dependency, especially the influence of global and local information of traffic flow sequences.

## Author contributions

**Conceptualization:** Anning Ji, Xintao Ma.

**Formal analysis:** Xintao Ma.

**Investigation:** Anning Ji, Xintao Ma.

**Methodology:** Anning Ji, Xintao Ma.

**Project administration:** Xintao Ma.

**Software:** Xintao Ma.

**Validation:** Xintao Ma.

**Writing – original draft:** Anning Ji.

**Writing – review & editing:** Xintao Ma.

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
