## [Decision Letter · Decision Letter 0]

14 Jan 2025

PONE-D-24-39892T-RippleGNN: Predicting traffic flow through Ripple propagation with attentive graph neural networksPLOS ONE

Dear Dr. Ma,

Thank you for submitting your manuscript to PLOS ONE. After careful consideration, we feel that it has merit but does not fully meet PLOS ONE’s publication criteria as it currently stands. Therefore, we invite you to submit a revised version of the manuscript that addresses the points raised during the review process.

We look forward to receiving your revised manuscript.

Kind regards,

Jinran Wu, PhD

Academic Editor

PLOS ONE

2. Please note that PLOS ONE has spec6ific guidelines on code sharing for submissions in which author-generated code underpins the findings in the manuscript. In these cases, all author-generated code must be made available without restrictions upon publication of the work. Please review our guidelines at https://journals.plos.org/plosone/s/materials-and-software-sharing#loc-sharing-code and ensure that your code is shared in a way that follows best practice and facilitates reproducibility and reuse.

3. Thank you for stating the following financial disclosure:  [Key Research Fund of Jilin Police College].

4. In the online submission form, you indicated that [Our data are only available upon request.].

5. Please ensure that you refer to Figures 5, 6 in your text as, if accepted, production will need this reference to link the reader to the figure.

Additional Editor Comments (if provided):

Reviewers' comments:

Reviewer's Responses to Questions

**Comments to the Author**

1. Is the manuscript technically sound, and do the data support the conclusions?

Reviewer #1: Yes

2. Has the statistical analysis been performed appropriately and rigorously? 

Reviewer #1: Yes

3. Have the authors made all data underlying the findings in their manuscript fully available?

Reviewer #1: Yes

4. Is the manuscript presented in an intelligible fashion and written in standard English?

Reviewer #1: Yes

5. Review Comments to the Author

Reviewer #1: The manuscript introduces a novel method for traffic flow prediction combining Ripple propagation with attentive graph neural networks. While the methodology is interesting and potentially impactful, some areas require clarification and tightening to enhance comprehensibility and academic rigor.

1. The abstract lacks precise quantification of the improvement over state-of-the-art methods. For instance, instead of ‘show the effectiveness of our approach’, provide concrete statistics.

2. The related work section briefly mentions GNNs but does not delve deeply into differences between T-RippleGNN and similar models like STFGNN and DyHSL. A more explicit comparison and explanation of advancements are necessary to contextualize the contribution.

3. The explanation of the Ripple module in Section 3.2 is overly dense and assumes a high level of familiarity with mathematical notations. Add an intuitive explanation or flowchart to complement the mathematical formulations, especially for concepts like Eq. (3) and Eq. (4).

4. The manuscript could benefit from a more transparent discussion of the limitations of T-RippleGNN. For example: What computational trade-offs exist for datasets with very large graph sizes? How does the choice of parameters (e.g., hop numbers) affect robustness?

5. Terms like "traffic profiles" and "traffic conditions" are used interchangeably. For clarity, standardize terms throughout the manuscript or explicitly define them early in the text.

6. While the analysis mentions hop numbers and embedding sizes, it does not explore how the choice of these parameters might generalize across different datasets. Consider including a discussion or additional experiments on parameter selection strategies for unseen scenarios.

7. The figures are informative but lack consistency in design (e.g., use of colors and legends). Ensure figures like Fig. 3 emphasize key trends or propagation mechanisms.

6. PLOS authors have the option to publish the peer review history of their article (what does this mean? ). If published, this will include your full peer review and any attached files.

**Do you want your identity to be public for this peer review?** For information about this choice, including consent withdrawal, please see our Privacy Policy .

Reviewer #1: No

---

## [Author Response · Author response to Decision Letter 1]

9 Mar 2025

Response to Reviewer 1 Comments

Dear Reviewers,

We greatly appreciate the constructive comments from all the reviewers and the opportunity we were given to improve our paper.

Following your comments and advice, we carried out new experiments and analyses to clarify unclear parts and adding more details to make the paper more complete.

Below we provide our response to your questions and concerns. Our response follows the following format:

Your comments

Response: our response and revision summary

“… quoted text of major content changes …”

We would like to thank you again for giving us this opportunity to improve the manuscript.

Yours Sincerely,

Authors.

1. The abstract lacks precise quantification of the improvement over state-of-the-art methods. For instance, instead of ‘show the effectiveness of our approach’, provide concrete statistics.

Response:

Thank you for your advices, we have added concrete statistics in the abstract part to illustrate the improvement as follows:

We evaluate our approach with three real-world traffic datasets. The results show that our approach reduces the prediction errors by approximately 2.24%-62,93% compared with state-of-the-art baselines and demonstrates the effectiveness of T-RippleGNN in traffic forecasting.

2. The related work section briefly mentions GNNs but does not delve deeply into differences between T-RippleGNN and similar models like STFGNN and DyHSL. A more explicit comparison and explanation of advancements are necessary to contextualize the contribution.

Response:

Thank you for your suggestions. It would improves the contribution to briefly states the differences. Thus we add to parts to give more explicit comparisons.

First, we add an additional explanation for DyHSL, which may suffer the problem of overfit and higher computational cost, and then compared with a representative method called STGNN:

STFGNN [35] enhances traffic forecasting by fusing temporal and spatial graphs with GCN and gated convolutions, with the capability to effectively deal with long sequence data. DyHSL [36] improves the forecasting with hypergraphs using HGNN for dynamics and interactive convolutions for spatial-temporal relations, which can capture higher-order interactions but at the cost of increased computational complexity and scalability issues. STGNN [40] integrates a novel graph neural layer, a recurrent layer and a transformer layer to capture the temporal and spatial dependency, which is more feasible for long period traffic speed prediction.

Then we modify the last paragraph of the related work, and explains the difference of our models with others, emphasizing the importance of Ripple propagation:

However, our framework differs from the above literatures that we utilize the propagation idea to capture the traffic network topology gradually, especially the heavy traffic states. We combine Ripple propagation with GRU in an iterative manner, where spatial propagation and temporal modeling are deeply intertwined. This enables T-RippleGNN to not only extract spatial features more effectively but also refine them iteratively based on temporal context, leading to a more comprehensive representation of traffic patterns. To be specific, both STFGNN and STGNN employs graph convolutional networks to model spatial dependencies, but its reliance on predefined static graphs limits its ability to adapt to dynamic traffic conditions and long-term prediction. In contrast, our model leverages Ripple to propagate traffic influences adaptively across multi-hop neighbors, capturing not only local spatial correlations but also global traffic patterns, especially under heavy traffic conditions.

3. The explanation of the Ripple module in Section 3.2 is overly dense and assumes a high level of familiarity with mathematical notations. Add an intuitive explanation or flowchart to complement the mathematical formulations, especially for concepts like Eq. (3) and Eq. (4).

Response:

Thank you for your suggestions. First, we move Figure 3 to line 219 between the Eq.3 and Eq.4, which illustrate the procedure of ripple propagation. This placement ensures that readers can directly reference the graphical illustration while engaging with the textual description of the propagation process, thereby reinforcing their understanding of key concepts such as multi-hop ripple sets, relevance probability calculation (Eq. 3), and iterative traffic profile aggregation (Eq. 4).

Second, we rewrite the caption of Figure 3 to better explain the procedure as follows:

The water wave shape of Ripple propagation procedure. The start points are randomly selected. The traffic profiles propagate to its immediate neighbors, forming the 1-hop ripple set (). During this step, the feature embeddings of the seed nodes are transmitted to their neighbors. The strength of this influence is represented by the intensity of the circle's color, with darker shades indicating stronger influence. This propagation continues hop-by-hop until the whole structure is propagated or the limit hop number is reached.

Third, we add further explanation for Eq.3 and Eq.4, which explains the effect of each equation:

(3)

where is the embedding dimension, and . This probability is calculated based on the similarity between the embeddings of the seed node and its neighbors, which reveals the impact strength between the connected traffic nodes.

Using the relevance probabilities, we aggregate the traffic profiles of the neighbors to compute the first hop response () for the seed node, as shown in Figure 3(b). This response represents the traffic state of the seed node after incorporating the influence of its immediate neighbors.

(4)

Thus, we can obtain the first response of the traffic condition in node n. The propagation continues iteratively for k-hops. At each step, the response from the previous hop () is used as the new seed to compute the next response (), by iteratively replacing by the previous response in Eq.3 with the node ripple sets for This allows the model to capture traffic influences from increasingly distant nodes, reflecting the dynamic and multi-scale nature of traffic networks.

4. The manuscript could benefit from a more transparent discussion of the limitations of T-RippleGNN. For example: What computational trade-offs exist for datasets with very large graph sizes? How does the choice of parameters (e.g., hop numbers) affect robustness?

Response:

Your advices are valuable to improve the parameter variation test. We have addressed how the hop number affects the result from line 377 to line 389. Additionally, we add a discussion to explain the computational trade-offs in the large graph sizes from line 402 to 412 as follows:

In summary, while T-RippleGNN demonstrates effective performance in traffic flow prediction, also has some limitations. The multi-hop propagation mechanism although effective for capturing spatial dependencies, incurs increased computational overhead when applied to datasets with extremely large graph sizes. Specifically, the iterative aggregation of h-hop neighbors leads to a time complexity proportional to , where is the number of edges. Thus, the choice of hyper parameters, particularly the maximum hop number , should be adapted according to the graph complexity. While larger values enable the capture of long-range spatial dependencies, they also introduce noise from irrelevant distant nodes and increase the risk of overfitting. Our ablation studies above reveal that performance plateaus when for most tested scenarios, reaching a balance between computational cost and efficiency.

5. Terms like "traffic profiles" and "traffic conditions" are used interchangeably. For clarity, standardize terms throughout the manuscript or explicitly define them early in the text.

Response:

Sorry for the fuzziness of the terms. We have change all the terms to “traffic conditions” for clarity, because they mean the same thing. We have given the definition of “traffic conditions” in section 3.2.

6. While the analysis mentions hop numbers and embedding sizes, it does not explore how the choice of these parameters might generalize across different datasets. Consider including a discussion or additional experiments on parameter selection strategies for unseen scenarios.

Response:

We sincerely thank you for raising this important point regarding the generalization of parameter choices across different datasets. We agree that a deeper discussion on parameter selection strategies is crucial for ensuring the robustness and applicability of T-RippleGNN in unseen scenarios. Based on the reviewer’s suggestion, we have expanded our discussion in the revised manuscript to address this issue. Below, we summarize the key additions:

In Parameter Variation, we have added a detailed discussion on how the optimal hop number varies across datasets.

In simpler graphs (e.g., SZ-Taxi), higher hop numbers () introduce less noise, as the propagation traces are less likely to overlap. However, in more complex graphs (e.g., PEMS-BAY), higher hop numbers lead to significant performance degradation due to overlapping propagation paths and increased noise. Thus in such cases, is a better parameter option.

Our experiments demonstrate that an embedding size of 32 achieves the best performance including large-scale graphs like PEMS-BAY. This consistency suggests that the chosen embedding size strikes a balance between expressiveness and efficiency, making it suitable for a wide range of scenarios.

This is because a size of 32 is enough to carry the information for propagating to neighbor nodes. In more generalized cases, the embedding size of 32, which achieves the best performance in the large scale datasets PEMS-BAY, strikes a balance between expressiveness and computational efficiency[15].

In future work, we propose exploring adaptive hop selection mechanisms that dynamically adjusting hyper parameters based on graph properties such as node degree distribution or network diameter.

We believe these additions can strengthen the manuscript by providing a more comprehensive discussion of parameter selection and generalization. We thank you for their valuable feedback, which has helped us improve the clarity and depth of our work.

7. The figures are informative but lack consistency in design (e.g., use of colors and legends). Ensure figures like Fig. 3 emphasize key trends or propagation mechanisms.

Response:

Sorry for the inconvenience. We have chose to use orange, green and purple to be main color for data expression. Thus we change our figure 1 and 2.

We also change the legends of Figure 3 to illustrate the propagation mechanisms to be clearer.

Fig. 3. The water wave shape of Ripple propagation procedure. The start points are randomly selected. The traffic conditions propagate to its immediate neighbors, forming the 1-hop ripple set (). During this step, the feature embeddings of the seed nodes are transmitted to their neighbors. The strength of this influence is represented by the intensity of the circle's color, with darker shades indicating stronger influence. This propagation continues hop-by-hop until the whole structure is propagated or the limit hop number is reached.

Thank you again for helping us improving the clarity and depth of our work

---

## [Decision Letter · Decision Letter 1]

26 Mar 2025

PONE-D-24-39892R1T-RippleGNN: Predicting traffic flow through Ripple propagation with attentive graph neural networksPLOS ONE

Dear Dr. Ma,

Thank you for submitting your manuscript to PLOS ONE. After careful consideration, we feel that it has merit but does not fully meet PLOS ONE’s publication criteria as it currently stands. Therefore, we invite you to submit a revised version of the manuscript that addresses the points raised during the review process.

We look forward to receiving your revised manuscript.

Kind regards,

Jinran Wu, PhD

Academic Editor

PLOS ONE

Reviewers' comments:

Reviewer's Responses to Questions

**Comments to the Author**

1. If the authors have adequately addressed your comments raised in a previous round of review and you feel that this manuscript is now acceptable for publication, you may indicate that here to bypass the “Comments to the Author” section, enter your conflict of interest statement in the “Confidential to Editor” section, and submit your "Accept" recommendation.

Reviewer #1: All comments have been addressed

2. Is the manuscript technically sound, and do the data support the conclusions?

Reviewer #1: Yes

3. Has the statistical analysis been performed appropriately and rigorously? 

Reviewer #1: Yes

4. Have the authors made all data underlying the findings in their manuscript fully available?

Reviewer #1: Yes

5. Is the manuscript presented in an intelligible fashion and written in standard English?

Reviewer #1: Yes

6. Review Comments to the Author

Reviewer #1: As noted in my previous review, the manuscript still contains issues with grammar and phrasing that affect clarity. While some improvements have been made, expressions like “traffic profiles can be floored through the edges” are unclear or incorrect. I recommend a thorough language revision to ensure technical accuracy and readability.

I suggest the authors consider citing the following recent works to enhance the related literature and better contextualize their contribution:

1. Multiscale-integrated deep learning approaches for short-term load forecasting

2. Temporal Graph Networks for Deep Learning on Dynamic Graphs

3. A hybrid robust system considering outliers for electric load series forecasting

7. PLOS authors have the option to publish the peer review history of their article (what does this mean? ). If published, this will include your full peer review and any attached files.

**Do you want your identity to be public for this peer review?** For information about this choice, including consent withdrawal, please see our Privacy Policy .

Reviewer #1: No

---

## [Author Response · Author response to Decision Letter 2]

11 Apr 2025

1. As noted in my previous review, the manuscript still contains issues with grammar and phrasing that affect clarity. While some improvements have been made, expressions like “traffic profiles can be floored through the edges” are unclear or incorrect. I recommend a thorough language revision to ensure technical accuracy and readability.

Response:

We apologize for the language problems in the manuscript. The language presentation was improved with assistance from a native English speaker with an appropriate research background. For example:

In this study, we argue that the traffic profiles can be propagated through the edges as the raindrops fall into the river and provoke ripples, in which multiple ripples superpose to form a resultant distribution of the traffic condition.

2. citing the following recent works to enhance the related literature and better contextualize their contribution.

Response:

Thank you for your suggestions. It would improves the related work for adding more literature.

First, we add the recommended literature in the reference part:

21. Y. Yang , Z Tao ,·Q Tao, Y Gao, J Wu. A hybrid robust system considering outliers for electric load series forecasting[J].Applied Intelligence, 2021.DOI:10.1007/s10489-021-02473-5.

29. E. Rossi , B Chamberlain, F Frasca. Temporal Graph Networks for Deep Learning on Dynamic Graphs[J]. 2020.DOI:10.48550/arXiv.2006.10637

30. Y. Yang, Y. Gao, Z. Wang, X. Li, H. Zhou, J. Wu. Multiscale-integrated deep learning approaches for short-term load forecasting. Int. J. Mach. Learn. & Cyber. 15, 6061–6076 (2024).

Then we contextualize the contribution in the relate work part:

Yang et al. proposed a hybrid model that integrated the extreme learning machine and an improved whale optimizer to enhance the robustness of the prediction [21].

TGN [29] provides a general framework that operates on dynamic graphs represented as event sequences and a novel training strategy that enables the model to learn from the sequential nature of the data while maintaining efficient parallel processing capabilities. Yang et al. pointed out that M-R-AR-BiGR [30] was combined with the traditional methods and bi-directional gate recurrent units to capture the temporal characteristics and improve the robustness.

---

## [Editor Report · Decision Letter 2]

15 Apr 2025

T-RippleGNN: Predicting traffic flow through Ripple propagation with attentive graph neural networks

PONE-D-24-39892R2

Dear Dr. Ma,

We’re pleased to inform you that your manuscript has been judged scientifically suitable for publication and will be formally accepted for publication once it meets all outstanding technical requirements.

Kind regards,

Jinran Wu, PhD

Academic Editor

PLOS ONE

---

## [Editor Report · Acceptance letter]

PONE-D-24-39892R2

PLOS ONE

Dear Dr. Ma,

I'm pleased to inform you that your manuscript has been deemed suitable for publication in PLOS ONE. Congratulations! Your manuscript is now being handed over to our production team.

Kind regards,

on behalf of

Dr. Jinran Wu

Academic Editor

PLOS ONE